# A multi-objective genetic algorithm to find active modules in multiplex biological networks

**Elva María Novoa-del-Toro** [1], **Efrén Mezura-Montes** [2], **Matthieu Vignes** [3], **Morgane Térézol** [1], **Frédérique Magdinier** [1], **Laurent Tichit** [4], **Anaïs Baudot** [1,5] *

**1** Aix Marseille Univ, INSERM, Marseille Medical Genetics (MMG), Marseille, France, **2** University of Veracruz, Artificial Intelligence Research Center, Veracruz, Mexico, **3** School of Fundamental Sciences, Massey University, Palmerston North, New Zealand, **4** Aix Marseille Univ, CNRS, Centrale Marseille, I2M UMR 7373, Marseille, France, **5** Barcelona Supercomputing Center, Barcelona, Spain

* anais.baudot@univ-amu.fr

**Data Availability Statement:** All relevant data are within the manuscript and its Supporting information files.

## Abstract

The identification of subnetworks of interest—or active modules—by integrating biological networks with molecular profiles is a key resource to inform on the processes perturbed in different cellular conditions. We here propose MOGAMUN, a Multi-Objective Genetic Algorithm to identify active modules in MUltiplex biological Networks. MOGAMUN optimizes both the density of interactions and the scores of the nodes (e.g., their differential expression). We compare MOGAMUN with state-of-the-art methods, representative of different algorithms dedicated to the identification of active modules in single networks. MOGAMUN identifies dense and high-scoring modules that are also easier to interpret. In addition, to our knowledge, MOGAMUN is the first method able to use multiplex networks. Multiplex networks are composed of different layers of physical and functional relationships between genes and proteins. Each layer is associated to its own meaning, topology, and biases; the multiplex framework allows exploiting this diversity of biological networks. We applied MOGAMUN to identify cellular processes perturbed in Facio-Scapulo-Humeral muscular Dystrophy, by integrating RNA-seq expression data with a multiplex biological network. We identified different active modules of interest, thereby providing new angles for investigating the pathomechanisms of this disease.

**Availability:** MOGAMUN is available at https://github.com/elvanov/MOGAMUN and as a Bioconductor package at https://bioconductor.org/packages/release/bioc/html/MOGAMUN.html.

**Contact:** anais.baudot@univ-amu.fr

## Author summary

Integrating different sources of biological information is a powerful way to uncover the functioning of biological systems. In network biology, in particular, integrating interaction data with expression profiles helps contextualizing the networks and identifying

**Funding:** EMNDT was supported by the Mexican National Council of Science and Technology (CONACYT); This research and MT salary were supported by the Excellence Initiative of Aix-Marseille University - A∗Midex, a French ''Investissements d'Avenir'' programme (https://www.univ-amu.fr/en/public/excellence-initiative) to AB. FM is supported by the TRIM-RD grant from AFM Telethon (https://www.afm-telethon.fr/). The funders had no role in study design, data collection and analysis, decision to publish, or preparation of the manuscript.

**Competing interests:** The authors have declared that no competing interests exist.

subnetworks of interest, aka active modules. We here propose MOGAMUN, a multi-objective genetic algorithm that optimizes both the overall deregulation and the density to identify active modules, considering jointly multiple sources of biological interactions. We demonstrate the performance of MOGAMUN over state-of-the-art methods, and illustrate its usefulness in unveiling perturbed biological processes in Facio-Scapulo-Humeral muscular Dystrophy.

This is a *PLOS Computational Biology* Methods paper.

## 1 Introduction

The success of functional genomics is associated with the massive production of quantitative information related to genes, proteins or other macromolecules. These data include, for instance, -omics molecular profiles measuring the expression or activity of thousands of genes/proteins, sensitivity scores resulting from RNA interference or CRISPR screenings, and GWAS scores providing significance of association between genes and phenotypic traits. These scores and measurements, often presented as *p*-values, intend to inform on the cellular responses associated to different cellular contexts. But transforming lists of deregulated genes/proteins and their associated *p*-values to sets of pathways and processes affected in the different cellular conditions remains a major challenge.

A classical approach to identify perturbed cellular processes is the search for over-representation of function or process annotations. Many tools exist that can take as input a list of genes, selected after defining a threshold for significance or ranked according to their *p*-values [1]. Such enrichment approaches will consider only the genes/proteins annotated in databases. Another set of successful approaches try to overcome this limitation by integrating scores or measures with biological networks. Biological networks are composed of nodes representing the biological macromolecules, often genes or proteins, and edges representing physical or functional interactions between those macromolecules. The goal is to identify *active modules*, i.e., subnetworks enriched in interactions and in nodes of interest. These active modules then facilitate the investigation of the perturbed cellular responses, as functional modules are the building blocks of cellular processes and pathways [2].

The identification of active modules from networks is an NP-hard problem [3–5]. Some active module identification algorithms are based on clustering co-expression networks [6, 7] or memetic algorithms [4]. However, most approaches rely on greedy searches, simulated annealing, and genetic algorithms (see [2] and [8] for general surveys of active module identification methods).

Algorithms based on greedy searches, such as PinnacleZ [9] and MATISSE [10], follow three general steps: i) selection of seed(s), ii) expansion of seed(s), and iii) significance test. In the selection of seed(s), a set of genes of interest (for instance, significantly differentially expressed genes) are picked. Then, the seed(s) are iteratively expanded (adding one node at a time), following a greedy criteria, i.e. choosing the node in the network neighborhood of the seed(s) that maximizes a score, which improves the module fitness. The expansion stops when any of the following three conditions is met: 1) the improvement of the score of the subnetwork is below a minimum threshold, 2) the subnetwork reached a maximum size, or 3) a

maximum distance from the seed(s) is reached. As a last step, the subnetworks are tested for significance, by comparing the score of each subnetwork with the score of a random subnetwork. These three steps are common to greedy searches algorithms, but every method has variations. For instance, the seeds selected by PinnacleZ are single nodes, whereas MATISSE selects connected subnetworks. The main drawback of greedy searches is that they can get trapped in local optima because at every step they only look at the local options. In particular, they cannot pick low scoring nodes, even if these can be key for escaping local optima and have access to several high scoring nodes.

Methods based on simulated annealing, such as jActiveModules [3], follow a hill-climbing philosophy, but instead of always picking the best option, i.e., the best neighbor node to be added to the subnetwork, they can also choose unfavorable options (i.e. options decreasing the global score), and thereby escape local optima. Algorithms based on simulated annealing follow two steps: i) initialization of nodes states, and ii) toggling of nodes states. In the initialization of nodes states, each node in the network gets either the active or inactive state, with a given probability. The set of active nodes constitutes the initial subnetwork, and the subnetwork's score is calculated as the aggregated score of its nodes. Then, in the second step, the nodes states are toggled: in every iteration, the state of a random node is changed from active to inactive, or vice versa. If the toggling improves the score of the subnetwork, it is always accepted; otherwise, it is accepted with a probability calculated based on the temperature parameter, which decreases gradually in every iteration. After toggling states for a given number of iterations, the highest scoring subnetwork found in any iteration is given as result. Some algorithms, such as jActiveModules, have a third step to evaluate the significance of the final subnetwork by comparing its score to scores obtained on randomized expression data. The main drawback of simulated annealing is that the bigger the input network is, the more iterations are needed in order to explore the full search space. Moreover, simulated annealing does not guarantee that the final set of nodes forms a single connected component. However, jActiveModules can filter such set of nodes, in order to keep the top-scoring single connected component(s).

Methods such as COSINE [11], the algorithm proposed by Muraro et al. [12], the one proposed by Ozisik et al. [13] or the one proposed by Chen et al. [5], are all based on genetic algorithms. A key feature of genetic algorithms is that several potential solutions are considered simultaneously. In a genetic algorithm, an initial population of individuals, i.e. subnetworks corresponding to potential solutions, is usually randomly generated. Each individual's fitness is then evaluated using one (mono-objective optimization) or several (multi-objective optimization) objective functions. The population of individuals then starts the evolution process, where new individuals are generated by crossing existing ones and by modifying them with mutations. The fittest individuals (those with better values for the objective function(s)) have a higher probability to be selected for the generation of offspring. The evolution stops when the algorithm converges, for instance, when there is no improvement in the best value for the objective function(s) for a given number of generations. One of the main advantages of genetic algorithms is that the crossover and mutation operators can help to find a balance between exploring different areas of the whole search space and exploiting the surroundings of promising regions. However, as in simulated annealing, standard crossover and mutation operators cannot guarantee that the final solution will have a set of nodes forming a single connected component. As an option, one can design customized crossover and mutation operators, as in [12, 13]. Importantly, genetic algorithms are capable of optimizing multiple (often conflicting) objectives simultaneously. If the problem is tackled as mono-objective, all the objectives are added into a single objective function by considering weights for each one of them, and the result is usually a single solution. In contrast, if the problem is defined as multi-objective, each

objective is associated with an independent objective function, and the result generally leads to several solutions that provide a trade-off for the values of the different objective functions.

By definition, active modules are expected to be enriched in interactions. However, to our knowledge, only few methods, such as SigMod [14], consider the density of interactions. Moreover, existing methods were designed for the analysis of single biological networks, usually a protein-protein interaction network. However, we now have access to several sources of physical and functional interactions between biological molecules. These interactions are represented in a diversity of biological networks, from networks encompassing metabolic and signaling pathways to networks representing correlation of expression. These different interaction networks, each having their own features, topology and biases, are better represented as multiplex networks. Multiplex networks are multilayer networks (i.e., networks composed of different layers, where every layer is an independent network), sharing the same set of nodes, but different types of edges [15]. We and others recently developed different approaches to study and leverage these more complex but richer biological networks [16–20]. In this work, we present MOGAMUN, a multi-objective genetic algorithm able to explore a multiplex network to identify several activez modules.

## 2 Materials and methods

### 2.1 The MOGAMUN algorithm

A multiplex network is defined as a triplet $G = <V, \mathbb{E}, C>$, where $V$ is the set of nodes, $\mathbb{E} = E_1, \ldots, E_\alpha$ correspond to the $\alpha$ different types of edges between the nodes in $V$, one type per layer of the multiplex network, and $C = \{(v, v, l, k): v \in V, l, k \in [1, \alpha], l \neq k\}$ is the set of coupling links that link every node $v$ with itself across the $\alpha$ layers. For every type of edge in a layer $l$, $E_l = \{(v_i, v_j): i \neq j, v_i, v_j \in V\}$ [21].

We introduce **MOGAMUN**, a Multi-Objective Genetic Algorithm to identify active modules from MUltiplex Networks. MOGAMUN is a customized version of the Non-dominated Sorting Genetic Algorithm II (NSGA-II) [22], adapted to deal with networks. NSGA-II is a widely used multi-objective genetic algorithm-based optimization method. NSGA-II uses non-dominated sorting to rank the solutions, which allows optimizing two objective functions simultaneously (see Supplementary Section 3 in S1 File. for details). Our goal is to identify subnetworks that jointly fulfil two objectives: the relevance of the nodes and the density of interactions, inside a given subnetwork.

**2.1.1 Definition of the objective functions.** We measure the relevance of the nodes in a subnetwork, using the **first objective function**, the average nodes score, defined in Eq (1).

$$\overline{NodesScore} = \frac{1}{n} \sum_{i=1}^{n} (Score_i^{norm}) \tag{1}$$

Where $n$ is the number of nodes in the subnetwork, and $Score_i = \Phi^{-1}(1 - p_i)$ is the weight of node $i$. $\Phi^{-1}$ is the inverse standard normal cumulative distribution function and $p_i$ is the resulting $p$-value, or False Discovery Rate (FDR)-corrected $p$-value, of a statistical test. A node is considered significant if its $p$-value/FDR is lower than a user-defined threshold. In many cases, it corresponds to the result of a differential expression analysis.

The calculus of the inverse normal cumulative distribution ($\Phi^{-1}$) leads to values in the range between $(-\infty, +\infty)$. We use Eq (2) to normalize the nodes scores to be in the [0, 1] range. The average nodes score $\overline{NodesScore}$ is thus also within this range. Notice that the average nodes score is not an aggregated z-score, as defined in [3], because our $Score_i^{norm}$ can be computed from either $p$-values or FDRs and is scaled to the range from 0 to 1. It is thereby not

necessarily distributed according to a standard normal distribution.

$$Score_i^{norm} = \frac{Score_i - min(Score)}{max(Score) - min(Score)} \qquad (2)$$

The **second objective function** intends to evaluate the density of interactions in a subnetwork. Here, we compute a normalized density in order to evaluate the density of a subnetwork in a multiplex network. We define the normalized density $D_{norm}$ in Eq (3).

$$D_{norm} = \sum_{l=1}^{L} \frac{d_s}{d_l} \qquad (3)$$

Where $L$ is the total number of layers in the multiplex network, $d_l$ is the overall density of layer $l$, and $d_s$ is the density of the subnetwork in layer $l$; the densities $d_s$ and $d_l$ are defined by Eq (4).

$$d = \frac{|E_t|}{|E_{max}|} \qquad (4)$$

Where $E_t$ is the total number of edges in $d_s$ or $d_l$, and $E_{max}$ is the number of edges of the complete graph of the corresponding size.

**2.1.2 General workflow of MOGAMUN.** We present the general flowchart of MOGA-MUN in Fig 1. We initialise the algorithm with a random population of individuals (parents). We then mate the initial population to create a new population (children) of the same size. Last, we select the best individuals out of the two populations (parents & children) to use them as parents in the next generation. We iteratively repeat the process until convergence. The step-by-step procedure is detailed below, in subsections 2.1.3 to 2.1.12. The algorithm parameters are presented in section 2.3.

We modified NSGA-II [22] to work with networks. To do so, we defined a coding scheme for the individuals with a variable length, where each feature corresponds to the identifier of a node. We also customized the original steps involving either the creation or the modification of individuals (generation of the initial population, crossover and mutation). In addition, we added a step to replace duplicated individuals with randomly generated ones, in order to ensure the diversity of the population and allow exploring the search space further. Importantly, we request all the individuals (i.e., the subnetworks of the multiplex network) to be single connected components.

**2.1.3 Generating the initial population.** We first defined a multiplex-network version of the Depth First Search (multiplex-DFS, see Algorithm 1), which allows generating individuals that are single connected components. In every iteration of the multiplex-DFS, a uniformly random layer of the multiplex network is visited (see Algorithm 1, line 9). We use the multiplex-DFS to generate an initial population of $N$ individuals. Each individual is a connected subnetwork with a random size between *MinS* and *MaxS*. The seed, i.e., the initial node in the network, is randomly chosen from the pool of significant nodes, in order to focus around interesting areas of the multiplex network.

**2.1.4 Evaluating the initial population.** We now evaluate all the individuals of the population, i.e., the set of potential subnetwork solutions, with the two objective functions described in Eqs (1) and (3). A high average nodes score implies that the individual contains high-scoring nodes. Similarly, a high normalized density implies that the individual is densely connected in the multiplex network.

**Algorithm 1** Multiplex Depth First Search

```
1: procedure DFS(M, seed)
2:    Let S be a stack
```

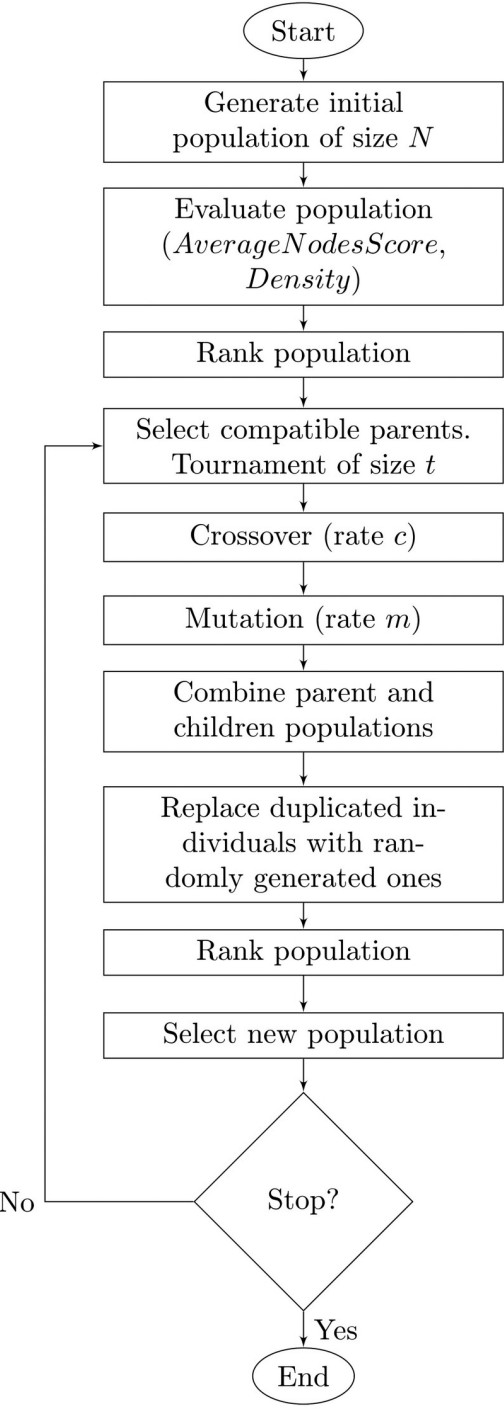

**Fig 1. General flowchart of MOGAMUN.**

```
3:   Let l be a layer from the multiplex network M
4:   S.push(seed)
5:   while S is not empty do
6:     v = S.pop()
7:     if v is not labeled as discovered then
```

```
8:       Label v as discovered
9:       l = pick a layer from M
10:       neighbors = get all direct neighbors of v in l
11:       neighbors = Shuffle(neighbors)
12:       for all i in neighbors do
13:         S.push(i)
14:       end for
15:     end if
16:   end while
17: end procedure
```

**2.1.5 Ranking individuals in the population.** We use the Pareto dominance, a classical criterion in evolutionary multi-objective optimization, to rank the individuals [23]. In a maximization problem, an individual $S1$ dominates $S2$ ($S1 \succ S2$), if $\overline{NodesScore}(S1) \geq \overline{NodesScore}(S2)$ and $D_{norm}(S1) \geq D_{norm}(S2)$, and at least one of the two inequalities is strict. The ranking process is carried out like in the original NSGA-II algorithm, as follows: initially, all non-dominated individuals (i.e., those individuals that are not dominated by any other individual in the current population) are assigned rank 1 and separated from the population. After that, from the remaining individuals in the population, those non-dominated are assigned rank 2 and separated from the population as well. Such process continues until there are no remaining individuals in the current population. At the end, all individuals in the population have a ranking value. The best individuals have rank 1.

Apart from assigning a rank to every individual, we also calculate their crowding distance, which is a measure that determines the proximity of the individuals in the objective space. The crowding distance of an individual is equivalent to the perimeter of the cuboid formed by its surrounding nearest pair of individuals in the same Pareto front, one at each side. The only exception is for those individuals that maximize one of the two objectives in each rank, which are directly assigned an infinite crowding distance value [23].

**2.1.6 Selecting compatible parents by tournament.** The parents are selected by tournament [24]. The selection of a pair of parents restricts the crossover to individuals that are compatible. This ensures that the children are also single connected components. Two individuals $S1$ and $S2$ are compatible if:

- $S1 \cap S2 \neq \emptyset$, or

- $S1 \cap \mathcal{N}(S2) \neq \emptyset$, where $\mathcal{N}(S2)$ is the set of neighbors of the nodes of $S2$.

The first parent is chosen via tournament (considering the rank of the individuals, and the crowding distance if they have the same rank). Depending on the number of compatible individuals, the second parent can be either selected also by tournament or directly assigned, if there is only one compatible individual. The procedure is described in Algorithm 2. If no individual is compatible with the first parent, we restart the process with a different individual as $Parent1$ (line 10 of Algorithm 2). If after a pre-specified number of attempts, the search of compatible parents is unsuccessful, we generate two random individuals, add them to the population of children and we skip crossover.

**Algorithm 2** Selection of compatible parents

```
1: Let Parent1 be the first parent, an individual selected from the
population, with a tournament of size t, based on ranking
2: Let CI be the list the compatible individuals with Parent1
3: Let N_CI be the number of compatible individuals with Parent1
4: if N_CI >= t then
5:   Let Parent2 be chosen via a tournament of size t from the individ-
uals in CI
6: else
```

```
 7:   if N_CI == 1 then
 8:     Let Parent2 be the only compatible individual
 9:   else
10:     if N_CI == 0 then
11:       Discard Parent1 and return to line 1
12:     end if
13:   end if
14: end if
```

**2.1.7 Crossover.** The goal of the crossover operator is to combine the nodes of two parent individuals, in an attempt to improve the values of any of the objective functions (average nodes score and/or normalized density). We mate the parents with crossover rate *c*. In order to guarantee that each child will be a single connected component, we use a crossover method inspired from the one proposed in Muraro et al., where the subnetworks corresponding to the parents are merged to have a single connected component [12]. In such a way, two nodes are randomly chosen, and two new children are generated with a Depth First Search, having as seed each selected node, respectively. However, our crossover varies according to two main aspects: 1) each seed for the children must correspond to significant nodes, and 2) the children can be generated either with Depth First Search or Breadth First Search. All children respect the subnetwork size's range.

**2.1.8 Mutation.** The goal of the mutation operator is to exploit the neighborhood of the children, adding/removing nodes, here also in an attempt to improve the value of any of the two objective functions. Notice that a node that is in the neighborhood of a child, i.e., directly connected to it, and that has a high node score, would allow increasing the average nodes score. In the same way, a neighbor node that is highly connected with the nodes of the child could improve the normalized density. We mutate each child independently with rate *m*. We first choose the list of potential nodes $v_p$ to be removed. We restrict this list to those nodes that can be removed without disconnecting the child subnetwork and that are not significant. We finalize the mutation process by adding $|v_p|$ new nodes to the subnetwork if $|v_p|>0$, or a single node if $|v_p| = 0$. The new nodes are chosen randomly, from the neighborhood of the corresponding child, considering all the layers of the multiplex network, and preferring significant nodes, if existing.

**2.1.9 Combining parent and children populations.** We join the parent and children populations, giving as result a population of size 2*N*.

**2.1.10 Replacing duplicated individuals with randomly generated ones.** Duplicates of individuals appear in the population when no compatible parents are found or when no crossover nor mutation are applied. To preserve diversity in the population, promote the exploration of the search space and avoid premature convergence, we introduce the replacement of duplicated individuals. To determine if an individual is duplicated, we check if it has more nodes in common with another individual than a given threshold $J_t$ of the Jaccard similarity coefficient. The Jaccard similarity coefficient *J* between two subnetworks (individuals) *A* and *B*, is calculated as follows:

$$J(A, B) = \frac{|A \cap B|}{|A \cup B|} = \frac{|A \cap B|}{|A| + |B| - |A \cap B|} \tag{5}$$

The dominated individual is labeled as duplicated. If no domination exists, one of the individuals is randomly selected.

**2.1.11 Selecting the new population.** After ranking the full population of size 2*N*, the new population of size *N* is selected with elitism. The top *N* ranked individuals will form the

new population, whereas the other $N$ are discarded. It is to note that the best individuals (among the parents and children) will thereby always remain in the population.

**2.1.12 Stopping criteria.** At this point, we have completed one generation. We iterate until reaching the stopping criteria, given by the number of generations *gen*. The result is the set of individuals in the **first Pareto front** (rank = 1). Evolutionary algorithms are stochastic search approaches, they must hence be run several times. As a result, we will have several first Pareto fronts. In order to select the final set of individuals, we calculate the *accumulated Pareto front*. To this goal, we take the results of all runs, re-rank them, and keep only those individuals in the new first Pareto front.

## 2.2 Post-processing the results

We designed a post-processing step to remove redundancy in the individuals obtained in the *accumulated Pareto front*. We calculated the Jaccard similarity coefficient (Eq 5) between every pair of individuals and merged them if it is higher than a given threshold $J_{t2}$.

## 2.3 Parameter values

In the study presented here, we used the parameter values listed in Table 1. We generated sub-networks in size range of [15–50], which corresponds to the size of communities identified by four over five top-performing algorithms in a community identification challenge [25]. This size range combined with a population of size 100 individuals, allows covering about a quarter of the multiplex network, around the most interesting areas. The minimum size parameter is important as real biological networks are sparse and the density definition will tend to favor smaller subnetworks. The final subnetworks will tend to have sizes equal to the minimum size allowed.

Tournament size and crossover rate are classical values in genetic algorithms [26–28]. Mutation rate of 10% is higher than in most approaches, to promote the exploitation of the search space near good solutions. It is to note that the algorithm converges with different combinations of crossover and mutation rates, and overall leads to very similar values of the final objective functions (Supplementary Section 4 in S1 File). We selected the total number of generations empirically, after running the algorithm several times in different contexts and monitoring its convergence. Similarly, both thresholds of the Jaccard similarity coefficient were also obtained empirically; we found that a relatively low value for the detection of duplicated individuals ($J_t$) allows to keep a high diversity rate, while preventing premature convergence, and a high value for the post-processing step ($J_{t2}$) allows merging individuals that vary only in a few number of nodes (i.e., merging individuals corresponding to very close subnetworks).

**Table 1. Parameters of MOGAMUN and values used in this study.**

| Feature | Description | Value |
|---------|-------------|-------|
| $N$ | Population size | 100 |
| $MinS$ | Minimum size of the individuals | 15 |
| $MaxS$ | Maximum size of the individuals | 50 |
| $t$ | Tournament size | 2 |
| $c$ | Crossover rate | 80% |
| $m$ | Mutation rate | 10% |
| $gen$ | Total number of generations | 500 |
| $J_t$ | Jaccard similarity coefficient threshold or duplicated individuals | 30% |
| $J_{t2}$ | Jaccard similarity coefficient threshold for the post-processing step | 70% |

## 2.4 Benchmark to compare the performance of MOGAMUN with existing methods

In order to compare the performance of MOGAMUN with state-of-the-art approaches, we used and extended the benchmark initially proposed by Batra et al. [29]. It is worth noticing that this benchmark works with a single interaction network as, to our knowledge, no active module identification method can consider multiplex networks. It artificially generates expression data to simulate a differentially expressed subnetwork. To this goal, the nodes of the network (i.e., the genes) are separated into two groups: Foreground Genes (FG) and Background Genes (BG). A *seed-and-select* algorithm is defined to randomly select the FG as a connected subnetwork. Such an algorithm selects a random seed and proceeds to iteratively add one node at a time to the subnetwork, by picking it up from the list of neighbors, such that the subnetwork remains connected. The process ends when the subnetwork reaches the desired size. Artificial expression data is then generated so that the FG contrasts with the BG, which means that the FG genes are artificially differentially expressed.

We computed the $F_1$ score (also known as $F$ score or $F$ measure) to evaluate the quality of the active modules identified by the different methods. The $F_1$ score is calculated on the union of all the active modules retrieved by each method over the 30 runs, using the Eq 6.

$$F_1 = 2 \times \left( \frac{precision \times recall}{precision + recall} \right) \tag{6}$$

Where $precision = \frac{TP}{TP+FP}$ and $recall = \frac{TP}{TP+FN}$. The $TP$ and the $FP$ are the number of FG and BG present in the active modules, respectively, and the $FN$ are the number of missing FG (i.e. the FG that were not retrieved).

**2.4.1 Benchmark networks.** We used two independent protein-protein interaction (PPI) networks in the benchmark (Table 2). *PPI_1* is the human protein reference database [30], taken from [29]. *PPI_2* was generated by merging interactions identified from several databases through the PSICQUIC portal [31] and the Center for Cancer Systems Biology (CCSB) Interactome database [32], taken from [16].

**2.4.2 Benchmark artificial expression data.** We simulated 2 different artificial expression datasets, one following a normal distribution and another one sampled from real RNA-seq data, as follows:

1. *Sim_normal.* We simulated expression data following a normal distribution. The mean values $\mu$ of the FG and BG groups of genes are $\mu(FG) = 5$, and $\mu(BG) = 2$, respectively, and a standard deviation SD = 1 for both groups of genes. This situation corresponds to a high signal strength, as in [29]. To test for differential expression, we performed a series of $t$-tests, and considered a gene as significantly differentially expressed if the $p$-value $\leq 0.05$. We used a set of 20 FG, and did not correct the $p$-value for multiple testing in order to make the test more challenging.

2. *Samp_TCGA.* We sampled data from real expression data, in order to have an RNA-seq distribution-like. We downloaded breast cancer RNA-seq expression dataset from The Cancer Genome Atlas breast cancer project (TCGA-BRCA) from the US National Cancer GDC

**Table 2. Interaction networks used in the benchmark.**

| Name | No. of nodes \|V\| | No. of edges \|E\| | Density $d$ |
|:---:|:---:|:---:|:---:|
| PPI_1 | 9425 | 36811 | $8.28 \times 10^{-4}$ |
| PPI_2 | 12621 | 66971 | $8.41 \times 10^{-4}$ |

**Table 3. Artificial expression datasets.**

| Name | Cases | Controls | Genes | Significant DE |
|---|---|---|---|---|
| *Sim_normal* | 100 | 10 | 9425 | 483 |
| *Samp_TCGA* | 1102 | 112 | 12621 | 20 |

portal (https://portal.gdc.cancer.gov/), as of May, 2019. This dataset is composed of 1 102 cases and 112 controls (we removed the outlier control sample "d5f0ea64.6660.49ac. a37e.3cd747045147"). To test for differential expression, we used the R package *edgeR* version 3.26.8 [33]. We consider a gene to be significantly differentially expressed if its False Discovery Rate FDR $\leq 0.05$ and $|log_2(FC)| > 1$, where FC (Fold Change) is the ratio of the difference in expression between cases/patients and controls. The expression data for the FG and BG were randomly sampled from the set of significantly differentially expressed and non-differentially expressed genes, respectively. We used a set of 20 FG.

In Table 3 we describe our two datasets. Columns "Cases" and "Controls" show the number of patients/cases and controls, respectively, "Genes" shows the total number of genes in the simulated dataset, corresponding to the total number of nodes in the networks, and "Significant DE" is the number of significantly differentially expressed genes.

**2.4.3 State-of-the-art algorithms selected for comparison.** We compared MOGAMUN with three selected methods, representative of the main approaches seeking for active modules: jActiveModules [3], PinnacleZ [9], and COSINE [11].

**jActiveModules.** Ideker et al. [3] proposed jActiveModules. They use a simulated annealing algorithm to find subnetworks with the highest scores, calculated from the differential expression of the subnetwork nodes. The search starts by selecting a subnetwork containing approximately half of the nodes of the full network. After that, they iteratively add or remove one node at a time from the selected subnetwork (the number of iterations is defined *a priori*). Whenever the addition or removal of a node increases the score of the subnetwork, the modification is accepted. Otherwise, it is accepted with a probability that decreases along the iterations, according to the temperature value. After finishing adding/removing nodes, the highest scoring subnetwork (found in any iteration) is selected as result. Finally, the significance of the selected subnetwork is evaluated. Several parameters can be tuned, but for the tests performed here, we used the default values recommended by the authors [3]. The only exception is that we set to 1 the number of modules to be retrieved, as this corresponds to the benchmark settings. jActiveModules is available as a Cytoscape plugin.

**COSINE.** Ma et al. [11] proposed COSINE, a method based on a standard mono-objective genetic algorithm. The goal of COSINE is to find the subnetwork with the highest change in expression among conditions, represented as node weights. COSINE further allows considering the level of co-expression between pairs of genes, represented as edge weights. To compute the edge weights, we calculated the co-expression of every pair of nodes connected in the benchmark networks. COSINE further allows giving more importance to either the weights of the nodes or the edges, with a parameter lambda. For the tests performed here, we used the same parameters as reported in [11], where COSINE is compared with other methods (number of iterations = 5000; zero to one ratio = 30), and we set lambda to 0.5, in order to give the same importance to changes in expression (i.e. node weights) and co-expressions (i.e. edge weights). COSINE is available as an R-package.

**PinnacleZ.** Chuang et al. [9] designed PinnacleZ, a greedy algorithm to identify active subnetworks that maximize the mutual information. The mutual information measures the differences in the distribution of the expression values of a given set of genes between two

conditions. PinnacleZ starts the search by selecting an initial set of seeds, and for each of these seeds, it iteratively adds the neighbor node that maximizes the mutual information of the subnetwork. The search stops when a maximal distance from the seed is reached or when the improvement of the mutual information score is not considered significant, given a threshold. PinnacleZ then performs three tests of significance on each of the identified active subnetworks, in order to guarantee that their individual mutual information is higher than the mutual information of a random subnetwork. We used the same parameters reported in [9] (distance from the seed = 2 nodes, minimal mutual information score improvement threshold = 0.05), and we set the maximum size per subnetwork = 50 (the same size that we allowed for MOGAMUN). PinnacleZ was originally available as a Java program and a Cytoscape plugin, but this latter one is no longer supported.

## 2.5 Application to Facio-Scapulo-Humeral muscular Dystrophy type 1 (FSHD1)

**2.5.1 RNA-seq expression data.** We used five Facio-Scapulo-Humeral muscular Dystrophy type 1 (FSHD1) RNA-sequencing expression datasets publicly available [34–36], extracted from the Gene Expression Omnibus [37]. We performed the differential expression analyses using the R package *edgeR* version 3.26.8 [33]. As recommended in the user guide of *edgeR*, we performed glmQLF tests for the two datasets with samples from different batches [35, 36], and Fisher Exact tests for the three datasets with samples from a single batch [34]. We considered a gene as a significantly Differentially Expressed Gene (significantly DEG) if the False Discovery Rate FDR $\leq 0.05$ and the $|log_2(FC)| > 1$, where the FC (Fold-Change) is the ratio of the difference in expression between cases and controls.

We selected from Yao et al. [34] RNA-seq data from muscle biopsies of 9 FSHD1 patients (quadriceps, 4 males and 5 females) and 9 controls (8 quadriceps, 1 tibialis anterior, 5 males and 4 females). We also selected data from two myoblasts derived from patients and the two corresponding myotubes, as well as two myoblasts from controls and three control myotubes (Supplementary Table S1 in S1 File). The cells were obtained from the University of Rochester repository and are described in Young et al. [38]. Our differential expression analyses revealed 6, 7 and 343 significantly DEGs, for biopsies, myoblasts and myotubes, respectively.

In Banerji et al. 2017 [35] RNA-sequencing was performed in triplicate on confluent immortalized myoblasts, for three FSHD1 patients (corresponding to 5 cell lines, for a total of 15 samples) and three healthy individuals (corresponding to 4 cell lines, for a total of 12 samples) (Supplementary Table S2 in S1 File). These cells were on one hand derived from a mosaic patient and described in Krom et al. [39] (54–12; 54–45; 54–2 for FSHD1 cells with 3 D4Z4 units and 54–6; 54-A10 as controls with 13 D4Z4 units). On the other hand, the 12Ubic and 16Ubic cells obtained from two FSHD1 patients and the 12Abic and 16ABic cells from matching controls are described in Homma et al. [40]. We identified 192 significantly DEGs comparing all the FSHD1 to control samples.

The last dataset was obtained from Banerji et al. 2019 [36] and corresponds to myotubes collected at the end of myoblasts to myotubes differentiation. These myotubes are derived from the myoblasts described in Banerji et al. 2017 [35]. In Banerji et al. 2019, a time course expression during differentiation was analyzed. We considered here only the last time point (T8) and selected triplicated samples for 5 FHSD1 patients and 4 controls (Supplementary Table S3 in S1 File). We identified 261 significantly DEGs.

**2.5.2 Biological interaction networks.** We built a multiplex network composed of three layers of physical and functional interactions (see Table 4). The nodes are either genes or proteins, considered here equally. The edges are undirected, and we removed loops (i.e.,

**Table 4. Multiplex biological network.**

| Name | No. of nodes $|V|$ | No. of edges $|E|$ | Density $d$ |
|---|---|---|---|
| PPI_2 | 12621 | 66971 | $8.41 \times 10^{-4}$ |
| Pathways | 10534 | 254766 | $4.59 \times 10^{-3}$ |
| Co-expression | 10458 | 1337347 | $2.45 \times 10^{-2}$ |

self-interactions). The three networks were taken from [16]. The first network (*PPI_2*), is a protein-protein interaction network, and is also the one used for the benchmark (Section 2.4). In the second network (*Pathways*), the links correspond to pathway interaction data, obtained with the R package *graphite* [41]. The last network (*Co-expression*), contains edges corresponding to correlations of expression. Spearman correlations were calculated from RNA-seq data of 32 tissues and 45 cell lines, and absolute correlations of at least 0.7 were selected to build the network [42].

## 3 Results

To the best of our knowledge, MOGAMUN is the first algorithm that detects active modules from multiplex networks. However, several methods exist to detect one [4, 10, 11], or several [3, 5, 6, 9, 12] active modules in monoplex networks -aka single networks. We here compare MOGAMUN with three state-of-the-art approaches to detect active modules in monoplex networks (section 3.1). We then applied our algorithm to study Facio-Scapulo-Humeral muscular Dystrophy type 1 (FSHD1), using a multiplex network (section 3.2).

### 3.1 MOGAMUN against state-of-the-art active module identification methods

We ran jActiveModules, COSINE, PinnacleZ and MOGAMUN 30 times (see Materials and methods). The execution times per run of each algorithm, in a desk computer with Intel processor i7 at 3.60GHz and 32GB of RAM, were approximately 30 min, 8 hours, 30 min, and 12 hours for jActiveModules, COSINE, PinnacleZ and MOGAMUN, respectively.

As a first experiment, we used the *PPI_1* network (Table 2) and the *Sim_normal* dataset (Table 3) (see Materials and methods). The goal is to retrieve the active module, which is a single subnetwork composed of 20 nodes (i.e., the foreground genes (FG)). The four methods retrieved the 20 nodes of the FG. PinnacleZ retrieved 13 231 subnetworks in total, corresponding to 494 subnetworks with at least one different node. These 494 subnetworks have an average size of 6 nodes and 3% average Jaccard similarity between them. COSINE and jActiveModules retrieved 30 subnetworks each, one per run. The 30 subnetworks found by COSINE all have at least one different node, an average size of 640 nodes and 5% average Jaccard similarity. Finally, 29 out of the 30 subnetworks retrieved by jActiveModules have at least one different node, an average size of 6 952 nodes and 76% average Jaccard similarity. MOGAMUN retrieved 6 modules with at least one different node, an average size of 17 nodes and 13% average Jaccard similarity. We calculated the $F_1$ score (Materials and Methods) of the union of all the active modules retrieved by each method on the 30 runs. The $F_1$ score determines how good the methods are to retrieve the foreground genes (FG) while avoiding picking background genes (BG). The $F_1$ score for jActiveModules, COSINE and PinnacleZ is close to zero ($F_1^{jActiveModules} = 0.00461$, $F_1^{COSINE} = 0.00507$, $F_1^{PinnacleZ} = 0.0319$), and over 0.4 for MOGAMUN ($F_1^{MOGAMUN} = 0.476$), as shown in Fig 2.

Overall, PinnacleZ identified the best modules in terms of both average nodes score and density (Fig 3A). However, the identified modules are small (average size of 6 nodes).

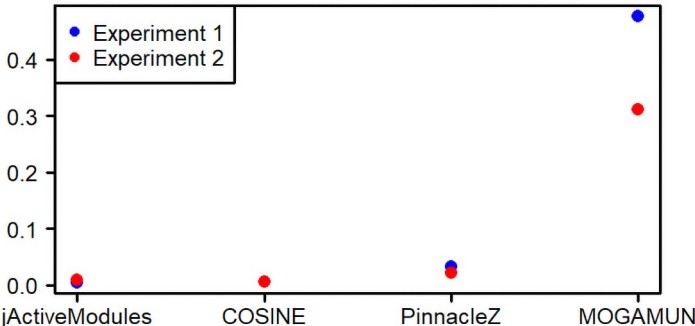

**Fig 2. $F_1$-scores obtained by jActiveModules, COSINE, PinnacleZ, and MOGAMUN in the two benchmark experiments.** Experiment 1 corresponds to *PPI_1* network and *Sim_normal* dataset; Experiment 2 corresponds to *PPI_2* and *Samp_TCGA* dataset.

PinnacleZ indeed restricts the search locally around the seed neighborhood by considering only the subset of nodes that are (at most) two steps away from the seed. In such small modules, the two objectives are expected to reach their maximum values. For example, a subnetwork composed only of two significant nodes with scores = 1 linked by an interaction will have a maximal average nodes score, as well as a maximal subnetwork density because it is a complete graph. In order to consider this, we filtered all the modules obtained by the four methods to keep only the subnetworks with at least 15 nodes (Fig 3B). This removed all the active modules obtained by PinnacleZ, and revealed that MOGAMUN succeeded to find the best results, in terms of the two objectives. MOGAMUN is the only approach allowing setting a minimum allowed size from the four methods implemented here. It is to note that, if we also remove the subnetworks with more than 50 nodes (the maximum size we set in MOGAMUN and PinnacleZ, the two methods allowing this setting), we would also discard all the results from COSINE and jActiveModules, which obtain subnetworks with hundreds or even thousands of nodes.

In a second test, we used the *PPI_2* network (Table 2) and the *Samp_TCGA* dataset (Table 3). The goal is also to retrieve a single active module, which is a subnetwork composed of 20 nodes (i.e., the foreground genes (FG), see Materials and methods). jActiveModules retrieved the highest number of FG genes (19/20), whereas PinnacleZ and MOGAMUN found 18/20 each, and COSINE, 12/20. The results are similar to the ones obtained in the first experiment (Fig 3C). PinnacleZ found 25 313 modules, out of which 1 055 have at least one different node. These 1 055 subnetworks have an average size of 6 nodes and 5% average Jaccard similarity between them. COSINE and jActiveModules retrieved 30 modules each, one per run. The 30 subnetworks found by COSINE all have at least one different node, with an average size of 205 nodes and 5% average Jaccard similarity. Four out of the 30 subnetworks retrieved by jActiveModules have at least one different node, with an average size of 1 033 nodes and 28% average Jaccard similarity between these 4 subnetworks. MOGAMUN retrieved 18 modules with at least one different node, an average size of 16 nodes and 18% average Jaccard similarity. The $F_1$ score of the union of all the active modules retrieved by each method on the 30 runs is below 0.1 for jActiveModules, COSINE and PinnacleZ ($F_1^{jActiveModules} = 0.00921$, $F_1^{COSINE} = 0.00601$, $F_1^{PinnacleZ} = 0.0211$), and over 0.3 for MOGAMUN ($F_1^{MOGAMUN} = 0.31$), as shown in Fig 2.

The filtering of the modules having more than 15 nodes in this comparison also removed all the results obtained by PinnacleZ, as well as two high-scoring subnetworks from jActiveModules (Fig 3D).

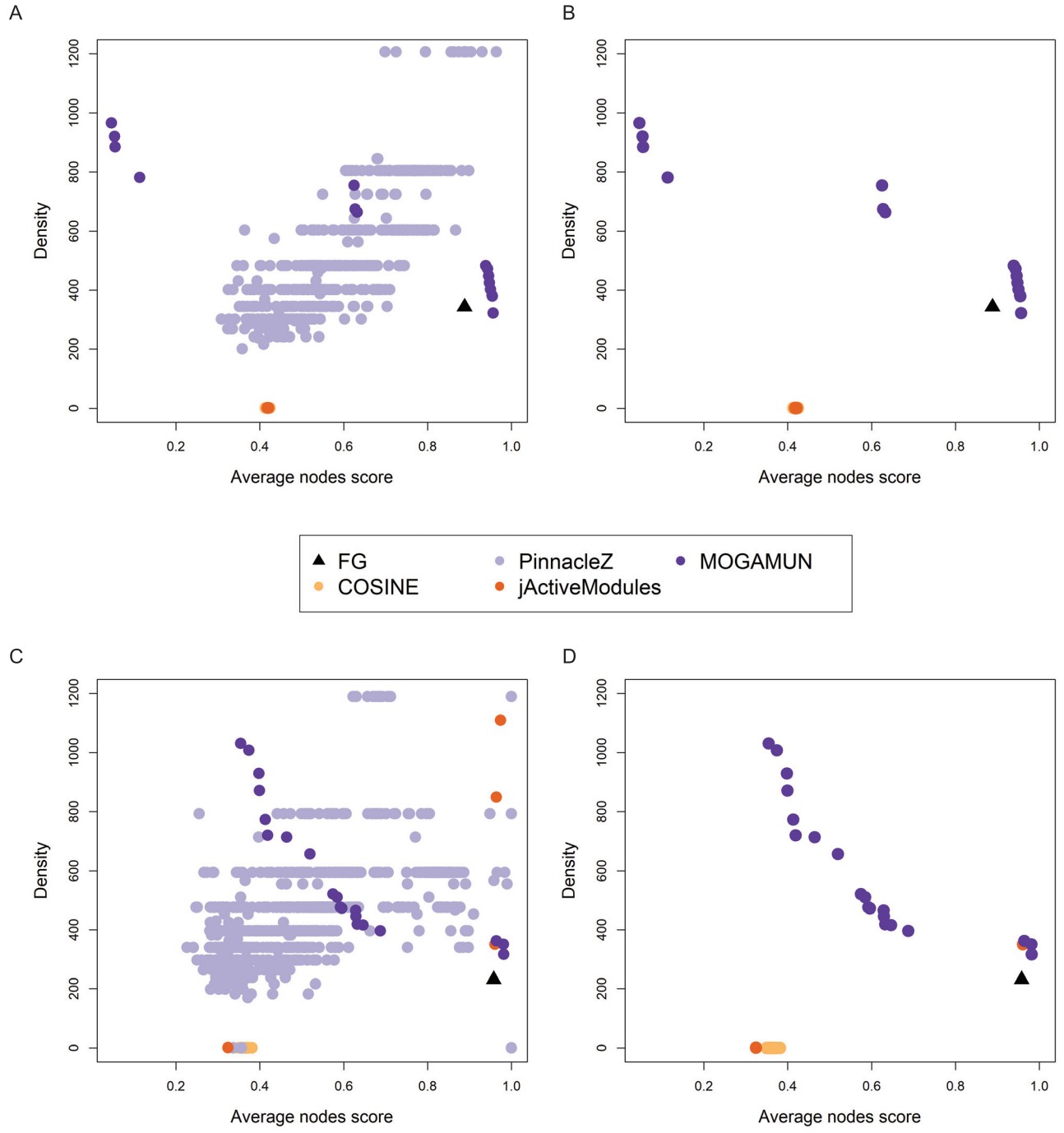

**Fig 3. Density and average node score of the subnetworks identified by MOGAMUN, COSINE, PinnacleZ and jActiveModules.** (A) Results of 30 runs using the *PPI_1* network and the *Sim_normal* dataset. (B) Filtered results from *(A)*, keeping only the subnetworks with at least 15 nodes. (C) Results of 30 runs using the *PPI_2* network and the *Samp_TCGA* dataset. (D) Filtered results from *(C)*, keeping only the subnetworks with at least 15 nodes. The size distributions of all the modules can be retrieved in Supplementary Figs S1-S8 in S1 File.

After filtering on module size, MOGAMUN led to the best results, although jActiveModules succeeded in finding a module with an average nodes score similar to one of MOGAMUN. However, MOGAMUN found overall denser subnetworks. This is expected as MOGAMUN is the only approach also optimizing subnetwork densities. If we also remove the subnetworks with more than 50 nodes, we would again discard all the results from COSINE and jActiveModules, with the exception of a single subnetwork, found by jActiveModules. In summary, MOGAMUN clearly identifies the best modules in terms of the multiple objective setting. Moreover, the retrieved modules have reasonable and tunable sizes.

## 3.2 Application to FSHD1

Facio-Scapulo-Humeral muscular Dystrophy type 1 (FSHD1) is a rare autosomal dominant genetic disease characterized by a progressive and asymmetric weakening of specific groups of muscles, with progression from the face to the lower limbs. The particularity of this disease resides in the absence of mutation in a gene encoding a muscle-specific factor. FSHD1 is however associated to a variable number of tandem repeats in the disease locus at the subtelomeric 4q35, more specifically to an array of 3.3 kb macrosatellite elements (D4Z4). In unaffected individuals, this array comprises between 11 and up to an average of 75 units [43]. In patients, this array is shortened with a threshold limit of less than 10 units. D4Z4 encodes the DUX4 transcription factor. The current pathological model associates D4Z4 array shortening with chromatin relaxation, expression of the DUX4 transcription factor and subsequent activation of a number of target genes of poorly known function in muscle physiology [44]. Overall, the biological processes leading to the muscle defects remain currently unclear.

We aim here to apply MOGAMUN in order to reveal biological processes that would not have been exposed by previous analyses, and further define biomarkers associated with the muscle phenotype of patients. We applied MOGAMUN using a multiplex network composed of three layers of biological interactions and FSHD1 RNA-sequencing expression datasets obtained from different types of cells [34–36] (Materials and Methods). More precisely, the first FSHD1 RNA-seq datasets were obtained from biopsies, myoblasts and myotubes differentiated from those myoblasts [34]. The two other datasets were obtained from immortalized myoblasts [35] and myotubes differentiated from those myoblasts [36] (Materials and Methods). We independently ran MOGAMUN 30 times.

We first considered the results obtained from [34] datasets, and analyzed the active modules identified by MOGAMUN in biopsies (18 active modules, Supplementary Fig S9 in S1 File), myoblasts (10 active modules, Supplementary Fig S10 in S1 File) and corresponding myotubes (23 active modules, Supplementary Fig S11 in S1 File). We also analyzed these three expression datasets with the other active module identification methods (jActiveModules, PinnacleZ, COSINE). Overall, these different active module identification approaches identified either extremely small or extremely large modules (see Supplementary Section 5 in S1 File for details on the results obtained by all the methods). MOGAMUN is the approach providing the best trade-off, as it allows retrieving modules that can be easily used for further biological interpretation.

In Yao's dataset, myoblasts, all the 10 active modules obtained by MOGAMUN contain at least one of the two significantly down-regulated genes LRRTM4 and GFRA1 (Supplementary Fig S10 in S1 File). LRRTM4 is required for presynaptic differentiation and GFRA1 belongs to the GDNF family receptor, also involved in the control of neuron survival and differentiation. The function of these two factors in muscle cells is, to our knowledge, not described. However, it is interesting to note that they belong to active modules containing proteins implicated in

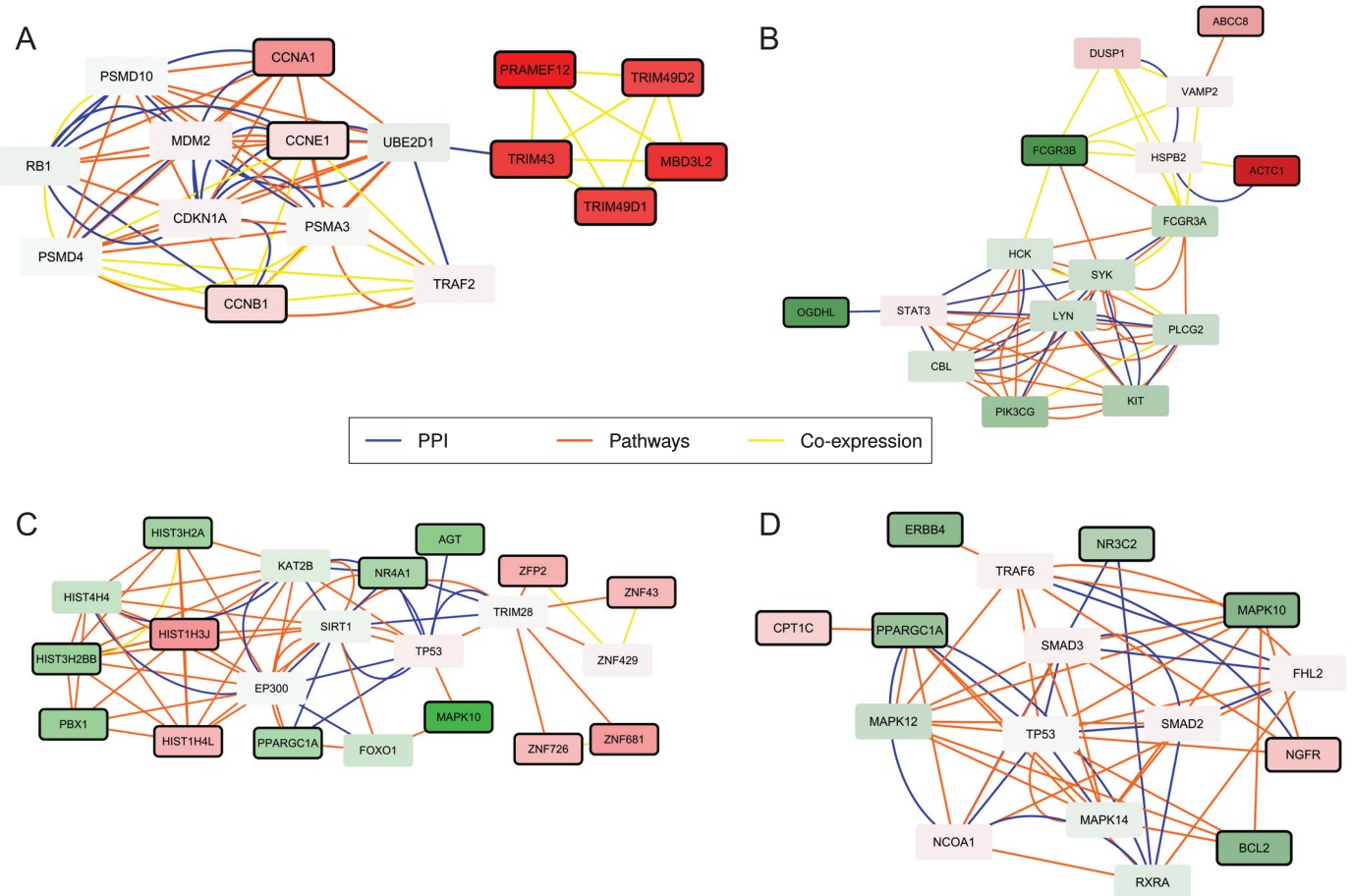

**Fig 4. Four active modules obtained by applying MOGAMUN on different FSHD1 expression datasets.** The color of the nodes represents the fold-change, where green and red nodes correspond to under- and over-expressed genes, respectively. Nodes with bold black border correspond to genes significantly differentially expressed (FDR <0.05 and absolute $log_2$ fold-change >1). Blue nodes correspond to genes with no associated transcriptomics data. The color of the edges represents the layer of the multiplex network, where blue, orange, and yellow correspond to PPI, Pathways, and Co-expression, respectively. The active modules are extracted from the sets of active modules obtained from (A) Yao's dataset, myotubes [34], (B) Yao's dataset, biopsies [34], (C) Banerji's 2017 dataset [35], and (D) Banerji's 2019 dataset [36].

ubiquitination, intracellular signaling and DNA replication machinery, as FSHD1 cells display increased apoptosis and reduced proliferation.

Using expression data obtained from myotubes differentiated from these myoblasts (Yao's dataset, myotubes), MOGAMUN identified 23 active modules, among which many contain a subset of highly over-expressed nodes (Supplementary Fig S11 in S1 File). These nodes associated to a high fold-change are DUX4 target genes, as defined in Geng et al. [44] and in Yao et al. [34] from DUX4-transduced over-expression experiments. In the active modules, the DUX4 target genes are however only linked together by interactions inferred from correlation of expression, and do not share pathway nor physical interactions. The over-expression of DUX4 target genes in differentiated cells is consistent with previous observations showing increased expression of this gene and its target genes upon differentiation. In some active modules, the DUX4 target genes are connected to cyclins through the Ubiquitin conjugating enzymes E2 D1 (Fig 4A). They are connected in particular to CCNA1, involved in cell cycle regulation at the G1/S and G2/M, and also reported as a DUX4 target gene in DUX4-transduced [34, 44] and immortalized [39, 45] myoblasts.

In biopsies, Yao et al. also detected some DUX4 targets genes at a low level [34]. However, we do not identify DUX4 target genes in active modules obtained by MOGAMUN from biopsy expression data.

Notably, in Yao's dataset, biopsies, we identified an interesting active module containing ACTC1 (encoding the Alpha Actin Cardiac Muscle 1) (Fig 4B). ACTC1 is mainly expressed in developing skeletal muscle, but its expression is also reactivated in diseased mature skeletal muscle, possibly as a sign of regeneration. In this module, we also noticed the presence of the over-expressed gene ABCC8, encoding a modulator of ATP-sensitive potassium channel and insulin release, and involved in the control of contractility and protection of the tissue against calcium overload and fiber damage. An intriguing observation is also the presence of OGDHL, encoding the 2-oxoglutarate dehydrogenase complex component E1-like, which localizes to mitochondria and degrades glucose and glutamate. OGDHL is significantly down-regulated in diseased biopsies. Overall, this active module links a potential reduced mitochondrial activity, frequently described for FSHD1, to an increased expression of factor involved in the calcium release when cellular energetics is compromised, and possible dysfunction of the contractile apparatus associated with over-expression of ACTC1, which might reveal an increased muscle regeneration or immaturity of the contractile apparatus.

In Banerji et al. 2017, the authors highlighted the repression of PAX7 target genes as a hallmark of FSHD1 skeletal muscle [35]. This signature, associated with the activation of the hypoxia pathway, was considered as more robust than the DUX4 signature, which remains variable between studies [34, 44, 46]. We identified 23 active modules with MOGAMUN using Banerji's 2017 dataset (Supplementary Fig S12 in S1 File). The nodes belonging to these modules, as observed for instance in Fig 4C, reveal MAPK-dependent decrease in cell signaling pathways, response to oxidative stress and reduced cell proliferation, as often reported for FSHD1 cells in culture.

We finally applied MOGAMUN to Banerji's 2019 dataset, which corresponds to RNA-seq data from myotubes derived from immortalized myoblasts [36]. This RNA-seq study was designed to consider the temporal dimension of gene expression. Genes are classified into 6 categories divided in 3 different groups: up or down regulated in FSHD1; up or down regulated during myogenesis and up or down regulated during FSHD1 myogenesis. One of the main message of this work relates to the suppression of PGC1$\alpha$ (encoded by the PPARGC1A gene) in FSHD1 myotubes as a cause of hypotrophy in FSHD1 myotubes [36]. We applied MOGAMUN only to RNA-seq data obtained from the last time point of the myoblast to myotubes differentiation kinetics (i.e. fully differentiated post-mitotic myotubes) (Materials and Methods). We identified 17 active modules (Supplementary Fig S13 in S1 File). An interesting module revealed, among other, connections between PPARGC1A and CPT1C (Fig 4D). PPARGC1, down-regulated in FSHD1, is involved in regulating the activities of cAMP response element binding protein (CREB) and nuclear respiratory factors (NRFs). CPT1C, up-regulated in FSHD, is involved in muscle glucose uptake. We also observed in the module the presence of MAPK10, required for protection against apoptosis. It is to note that MAPK10 is also identified in a module from FSHD1 immortalized myoblasts Fig 4C), overall highlighting the existence of connections between specific signalling pathways and chromatin-associated factors previously identified as implicated in the disease (YY1, EP300, CREBBP) [46, 47].

## 4 Discussion

We here designed, compared and applied MOGAMUN, a multi-objective genetic algorithm that is able to detect active modules in multiplex networks. Multiplex biological networks are composed of different layers of physical and functional interactions; each layer has its own

meaning, topology and noise. The protein-protein interaction layer, for example, is sparse, but composed of physical binary interactions extracted from curated databases. On the other hand, the co-expression network is very dense, but prone to indirect and spurious interactions. However, altogether, the different layers can provide complementary functional information [16, 17].

We compared MOGAMUN to three different methods, representative of the main algorithms dedicated to the identification of active modules: greedy searches (PinnacleZ), simulated annealing (jActiveModules), and mono-objective genetic algorithm (COSINE). As, to our knowledge, no existing method is able to leverage multiplex networks as inputs, we designed a benchmark for comparison that is based on single networks. In order to have a fair comparison between MOGAMUN and the three other methods, we used the parameters recommended by the authors, except when parameter values could be tuned to match the benchmark scenario or the parameters selected for MOGAMUN (Table 1). In particular, we set the number of subnetworks to be retrieved by jActiveModules to one in the benchmark because there was a single active module. In addition, we set the maximum size per subnetwork to 50 nodes in PinnacleZ. Finally, in COSINE, we set the lambda parameter to 0.5, to have a trade-off between the weights of the nodes and edges. Both the weights of the nodes and edges in COSINE are calculated from the expression data (section 2.4.3), and they cannot be manipulated by the user. Importantly, none of the four methods compared in this manuscript can handle weighted networks.

In the benchmark analyses, we observed that jActiveModules and COSINE tend to retrieve very large subnetworks (up to hundreds or even thousands of nodes), whereas PinnacleZ tends to retrieve very small ones (mostly composed of 2 or 3 nodes). Although jActiveModules, COSINE and PinnacleZ were able to retrieve most or all of the simulated foreground genes, their $F_1$ scores were lower than those of MOGAMUN because they selected more background genes (i.e., false positives). Importantly, only PinnacleZ and MOGAMUN have a user-defined threshold to limit the maximum size of the modules, and MOGAMUN is the only method having a user-defined threshold to limit the minimum size of the modules.

In addition to the benchmark with artificial expression data, we also applied the four methods to the three RNA-Seq datasets of FSHD1 from [34]. MOGAMUN is the only method that can take as input more than one network. Hence, in order to apply all the approaches, we aggregated the three network layers of the multiplex biological network (Table 4). The behaviour of jActiveModules, COSINE and PinnacleZ is similar in this real-case experiment as in the benchmark analyses (see Supplementary Section 5 in S1 File for detailed results). jActiveModules and COSINE tend to retrieve large subnetworks, and PinnacleZ finds only very small ones (Supplementary Figs S17-S25 in S1 File). We further observed that, as expected, the simple density (Eq 4) is higher for MOGAMUN active modules obtained from the aggregated networks, whereas the normalized density (Eq 3) is higher for MOGAMUN active modules obtained from the multiplex network (Supplementary Figs S17-S25 in S1 File). In addition, since the aggregated network is denser than any of the individual network layers, the chances of connecting high-scoring nodes is also higher. In the future, a benchmark based on biological proxies, using for instance Gene Ontology annotation enrichments [48], as recently proposed in the DOMINO approach [49], could be used to compare the active modules obtained from aggregated versus multiplex networks.

We demonstrated the performance of MOGAMUN in retrieving modules both densely connected and containing top-scoring nodes, i.e., nodes associated with a high deregulation. The two objective functions of MOGAMUN overall allow using an aggregated node score to optimize the differential expression of the subnetworks while balancing their size. However, MOGAMUN tends to find subnetworks of the minimum size allowed (or very close to it).

This is expected, given the sparse nature of biological networks. This bias towards minimum size active modules could lead to having a large active module cut into several smaller subnetworks, which are nonetheless easier to visualize and interpret. Overall, the *MinS* parameter has to be chosen carefully, depending on the user's needs. We implemented in addition a post-processing step to merge overlapping subnetworks, according to a user-defined threshold. It is to note that, given the small-world property of biological networks, if this threshold is too low, the result might be a single (large) active module. We therefore recommend trying different thresholds values.

MOGAMUN running time is, similarly to the other genetic algorithm COSINE, one order of magnitude slower than jActiveModules and PinnacleZ in its current implementation. This running time could be improved by implementing the most computationally demanding tasks (e.g., crossover) in lower programming languages, like C or C++, or using surrogate-assisted multi-objective evolutionary algorithms.

The extensive analysis of the different FSHD1 datasets highlighted the reduced proliferation and increased apoptosis of cells from FSHD1 patients and led to the identification of novel genes in these different pathways by linking cell defects to factors involved in muscle function. It further revealed consistencies in biological processes identified by different teams in their respective models but also some putative discrepancies in the interpretation of disease-associated biological processes depending on the type of samples used (biopsies of muscle unaffected in the disease, immortalized or transduced proliferative myoblasts or post-mitotic myotubes). Overall, this also reveals that MOGAMUN can be applied to identify disease-associated biological processes in rare diseases for which the number of samples is limited, and also to compare the processes identified in different datasets.

We applied here MOGAMUN to identify active module from the integration of RNA-seq expression data into multiplex networks. However, it is to note that any type of molecular profile associated to *p*-values can be integrated on the networks, such as *p*-values obtained from a GWAS, from phenotypic hit screening, or from proteomics profiling.

## Supporting information

**S1 File. Supporting text, tables and figures. 1. Supplementary Figs:** Fig S1: Sizes of the subnetworks identified by PinnacleZ in the experiment using the network *PPI_1* and the simulated data with normal distribution. Fig S2: Sizes of the subnetworks identified by PinnacleZ in the experiment using the network *PPI_2* and the sampled data from RNA-Seq TCGA breast cancer dataset. Fig S3: Sizes of the subnetworks identified by COSINE in the experiment using the network *PPI_1* and the simulated data with normal distribution. Fig S4: Sizes of the subnetworks identified by COSINE in the experiment using the network *PPI_2* and the sampled data from RNA-Seq TCGA breast cancer dataset. Fig S5: Sizes of the subnetworks identified by jActiveModules in the experiment using the network *PPI_1* and the simulated data with normal distribution. Fig S6: Sizes of the subnetworks identified by jActiveModules in the experiment using the network *PPI_2* and the sampled data from RNA-Seq TCGA breast cancer dataset. Fig S7: Sizes of the subnetworks identified by all the methods in the experiment using the network *PPI_1* and the simulated data with normal distribution. Fig S8: Sizes of the subnetworks identified by all the methods in the experiment using the network *PPI_2* and the sampled data from RNA-Seq TCGA breast cancer dataset. Fig S9: Yao's dataset, biopsies: Active modules 1–18. Fig S10: Yao's dataset, myoblasts: Active modules 1–10. Fig S11: Yao's dataset, myotubes: Active modules 1–23. Fig S12: Banerji's 2017 dataset: Active modules 1–23. Fig S13: Banerji's 2019 dataset: Active modules

1–17. **2. Supplementary Tables.** Table S1: Samples from Yao's datasets. Downloaded from https://www.ncbi.nlm.nih.gov/geo/query/acc.cgi?acc=GSE56787. Table S2: Samples from Banerji's 2017 dataset. Downloaded from https://www.ncbi.nlm.nih.gov/geo/query/acc.cgi?acc=GSE102812. Table S3: Samples from Banerji's 2019 dataset. Downloaded from https://www.ncbi.nlm.nih.gov/geo/query/acc.cgi?acc=GSE123468. **3. Non-dominated Sorting Genetic Algorithm II (NSGA-II).** Algorithm S1: Fast non-dominated sorting. Fig S14: Crowding distance concept. Algorithm S2: Crowding distance assignment. Algorithm S3: Elitist selection of a new population. **4. MOGAMUN Genetic Algorithm parameter tuning.** Fig S15: Convergence plots of the average nodes score (A) and density (B). At each generation, the best values for the average nodes score and density of the 30 runs are averaged and plotted. Fig S16: Average nodes score (A), density (B), and overlapping nodes (C) of the active modules obtained in the *accumulated Pareto fronts* of 30 runs of MOGAMUN with different combinations of parameters. **5. Application to Facio-Scapulo-Humeral muscular Dystrophy type 1 (FSHD1).** Fig S17: Sizes of the subnetworks identified by the five approaches on Yao's dataset, biopsies. Fig S18: Size, average nodes score and density of the subnetworks obtained by the different methods using Yao's dataset, biopsies. The sizes of the subnetworks are represented on a log scale. The density is computed either on the aggregated network, corresponding to the union of the three biological networks used in this study, or using the multiplex-normalized density, proposed in the main manuscript. Fig S19: Size, average nodes score and density of the subnetworks obtained by the different methods using Yao's dataset, biopsies, selecting only the subnetworks containing at least 15 nodes. The sizes of the subnetworks are represented on a log scale. The density is computed either on the aggregated network, corresponding to the union of the three biological networks used in this study, or using the multiplex-normalized density, proposed in the main manuscript. Fig S20: Sizes of the subnetworks identified by the five approaches on Yao's dataset, myotubes. Fig S21: Size, average nodes score and density of the subnetworks obtained by the different methods using Yao's dataset, myotubes. The sizes of the subnetworks are represented on a log scale. The density is computed either on the aggregated network, corresponding to the union of the three biological networks used in this study, or using the multiplex-normalized density, proposed in the main manuscript. Fig S22: Size, average nodes score and density of the modules obtained by the different methods using Yao's dataset, myotubes, selecting only the subnetworks containing at least 15 nodes. The sizes of the subnetworks are represented on a log scale. The density is computed either on the aggregated network, corresponding to the union of the three biological networks used in this study, or using the multiplex-normalized density, proposed in the main manuscript. Fig S23: Sizes of the subnetworks identified by the five approaches on Yao's dataset, myoblasts. Fig S24: Size, average nodes score and density of the subnetworks obtained by the different methods, using Yao's dataset, myoblasts. The sizes of the subnetworks are represented on a log scale. The density is computed either on the aggregated network, corresponding to the union of the three biological networks used in this study, or using the multiplex-normalized density, proposed in the main manuscript. Fig S25: Size, average nodes score and density of the subnetworks obtained by the different methods, using Yao's dataset, myoblasts, selecting only the modules containing at least 15 nodes. The sizes of the subnetworks are represented on a log scale. The density is computed either on the aggregated network, corresponding to the union of the three biological networks used in this study, or using the multiplex-normalized density, proposed in the main manuscript. Table S4: Number of genes and number and percentage of differentially expressed genes (DEGs) retrieved in the active modules by the different methods in 30 runs. Table S5: Number of genes and number and percentage of differentially expressed genes (DEGs) retrieved in the active modules by the different

methods in 30 runs. Only the statistics of the subnetworks with at least 15 nodes are reported here.
(PDF)

## Acknowledgments

The bioinformatics analyses were performed on the Core Cluster of the Institut Français de Bioinformatique (IFB) (ANR-11-INBS-0013), on the Centre de Calcul Intensif d'Aix-Marseille, and on the Genotoul cluster. We thank Steffen Neuman for his help for the submission of MOGAMUN as a Bioconductor package, and Elisabeth Remy (Aix Marseille Univ, CNRS, I2M, Marseille Institute of Mathematics, Marseille, France), and Claude Pasquier (Univ. Nice Sophia Antipolis, CNRS, I3S, Nice, France), for useful discussions.

## Author Contributions

**Conceptualization:** Elva María Novoa-del-Toro, Efrén Mezura-Montes, Laurent Tichit, Anaïs Baudot.

**Data curation:** Elva María Novoa-del-Toro, Morgane Térézol.

**Formal analysis:** Elva María Novoa-del-Toro, Morgane Térézol, Anaïs Baudot.

**Funding acquisition:** Anaïs Baudot.

**Investigation:** Elva María Novoa-del-Toro, Efrén Mezura-Montes, Frédérique Magdinier, Anaïs Baudot.

**Methodology:** Elva María Novoa-del-Toro, Efrén Mezura-Montes, Matthieu Vignes, Laurent Tichit, Anaïs Baudot.

**Project administration:** Anaïs Baudot.

**Resources:** Frédérique Magdinier, Anaïs Baudot.

**Software:** Elva María Novoa-del-Toro, Laurent Tichit.

**Supervision:** Efrén Mezura-Montes, Matthieu Vignes, Laurent Tichit, Anaïs Baudot.

**Validation:** Elva María Novoa-del-Toro, Morgane Térézol, Frédérique Magdinier.

**Visualization:** Elva María Novoa-del-Toro, Morgane Térézol.

**Writing – original draft:** Elva María Novoa-del-Toro, Anaïs Baudot.

**Writing – review & editing:** Elva María Novoa-del-Toro, Efrén Mezura-Montes, Matthieu Vignes, Frédérique Magdinier, Laurent Tichit, Anaïs Baudot.

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
