## [Decision Letter · Decision Letter 0]

5 Jan 2021

Dear Dr Baudot,

Thank you very much for submitting your manuscript "A Multi-Objective Genetic Algorithm to Find Active Modules in Multiplex Biological Networks" for consideration at PLOS Computational Biology.

As with all papers reviewed by the journal, your manuscript was reviewed by members of the editorial board and by several independent reviewers. In light of the reviews (below this email), we would like to invite the resubmission of a significantly-revised version that takes into account the reviewers' comments.

As a Methods Research Article, testing and validation of your algorithm is critical. A revision will need to address the comments raised by Reviewer 3, especially the concerns about benchmark networks used to evaluate MOGAMUN. Please feel free to contact me if you would like to discuss your revision plan.

We cannot make any decision about publication until we have seen the revised manuscript and your response to the reviewers' comments. Your revised manuscript is also likely to be sent to reviewers for further evaluation.

Sincerely,

Paul Jensen

Guest Editor

PLOS Computational Biology

Mona Singh

Methods Editor

PLOS Computational Biology

As a Methods Research Article, testing and validation of your algorithm is critical. A revision will need to address the comments raised by Reviewer 3, especially the concerns about benchmark networks used to evaluate MOGAMUN. Please feel free to contact me if you would like to discuss your revision plan.

Reviewer's Responses to Questions

**Comments to the Authors:**

Reviewer #1: The work claims to present a first of its kind way to identify active sub-networks in a network where there are multiple types of edges-- multiplex networks:

There is a disconnect between the paragraphs where the explanation of Equation (4) ends and the explanation of the flowchart begins.

There needs to be a brief overview of what exactly NSG-II does, and why it is possible and viable to adapt it to work with multiplex networks.

There are contractions, expansions of which are nowhere in the text. The first usage of a contraction needs to have the full name and explanation.

Another work, a brief evaluation of which should be in this work: DIRAC: Identifying Tightly Regulated and Variably Expressed Networks by Differential Rank Conservation (DIRAC) by James A. Eddy, Leroy Hood, Nathan D. Price, and Donald Geman, published in PLOS Computational Biology in 2010.

Were tests done to explore the application to FSHD1, using the three methods for comparison: jActive, PinnacleZ and COSINE? If yes, how do those compare with MOGAMUN?

Overall, the paper is well-written and is easy to follow.

Reviewer #2: The paper deals with a multi-objective optimization and presents a new algorithm, called MOGAMUN, for searching active modules in multiplex networks.

Line 267: "We elected" (We selected??)

I suggest to merge subsections 2.2.3 - 2.2.6 into one subsection.

Put the caption in Table 4 on the top of the table.

Reviewer #3: In this manuscript, the authors proposed MOGAMUN, a multi-objective genetic algorithm to find active modules in multilayer networks. The initial population is generated from a set of selected genes such as differentially expressed genes, then in each step optimization is done using two objective functions that optimize node scores and density of subnetworks. Node scores are defined based on p-values obtained mainly from differential expression analysis. The proposed method is evaluated by comparing against three other methods on simulated single-layer networks.

General comments:

1. Authors claim “MOGAMUN is the first method able to use multiplex networks”. This is not an accurate claim. There are previous studies that have already covered this topic. Several such references are given below in the specific comments section. Therefore, it needs to be mentioned clearly what are the novel aspects of this work.

2. Using density of subnetworks does not seem to be a good choice for objective function. Based on definition of objective function, a group of small dense subnetworks of a module may have higher score than the actual module. As a result, the algorithm might get stuck in local optima split the actual module to smaller submodules. A module should satisfy both conditions: i) dense connections among nodes in the module; ii) fewer connections with other nodes in the graph. However, in this work it seems like little attention is paid to the second condition.

3. The evaluation of the proposed method is not rigorous. For example, comparison with other methods is based on two networks with only one subnetwork. More realistic datasets with multiple active modules are needed. In addition, in Figure 2, methods are compared based on density and node scores which are the metrics that have been specifically optimized using MOGAMUN, and therefore not a fair comparison. More importantly, since this method is dedicated to work on multiplex networks the comparison should also be on multiplex network, not on single network.

Specific comments:

1. The results of “community identification dream challenge” showed that dedicated methods for complex network fail to significantly improve the task of module discovery compared to methods that considered each network individually. I am interested to see how your method works on multiplex networks compared to methods that are designed to work on single networks. Two possible strategies for single network methods to be applied on multiplex networks are merging networks before module discovery or consensus analysis after applying method on individual network. In fact, the second objective function seems to be equivalent to pre-merging the networks after normalizing the edge weights in each network by the overall network density. With the way that the networks are simulated, it may be easy to simply compute differential expression, and then looking for densely connected subnetworks from the top DE genes.

2. What is the running time of MOGAMUN and how does it compare to other methods?

3. COSINE is another method based on genetic algorithm. In evaluation section some discussion on why MOGAMUN outperforms COSINE should be added.

4. How the hyper parameters were tuned? Given the small dataset tested, it is unclear how these set of parameters can be selected for other networks with different structures and topology.

5. How close different layers of the networks should be? Should they come from same context and same tissue? It seems the relation of different layers may have a big impact on the performance of the method. Do you have any elaboration on this?

6. Some critical details are missing in the paper. For example, in section 2.3.2 describes how the expression levels for the FG genes are sampled, but did not describe how the FG genes are selected in the first place. Are they random genes in the PPI network? Most connected ones? The choices of the FG genes will directly affect the ability of the algorithms to find the module. For TCGA data, it is also confusing that you have to “sample” some gene expression data while there is (almost) a one to one correspondence between the PPI network and the RNAseq data.

7. For general audience not familiar with genetic algorithms, it would also be helpful to add some details about tournament size, and crossover rate and how these parameter values impact the algorithm’s behavior.

8. The definition of multiplex network provided by the authors includes C, coupling links between layers. This is not explained, and seems to have not been used in the paper at all.

9. Table 3, why for sim_normal there are 483 DE genes? This suggests multiple testing correction is needed to pick the DE genes.

10. Figure 2 has very poor quality.

11. Some references for module discovery in multiplex networks:

Haiyan Hu, Xifeng Yan, Yu Huang, Jiawei Han and Xianghong Jasmine Zhou, Mining coherent dense subgraphs across massive biological networks for functional discovery, Bioinformatics 2005.

Yu, L., Yao, S., Gao, L., & Zha, Y. (2019). Conserved Disease Modules Extracted From Multilayer Heterogeneous Disease and Gene Networks for Understanding Disease Mechanisms and Predicting Disease Treatments. Frontiers in genetics, 9, 745.

Huang, X., Chen, D., Ren, T. et al. A survey of community detection methods in multilayer networks. Data Min Knowl Disc (2020). https://doi.org/10.1007/s10618-020-00716-6

De Domenico, M., Lancichinetti, A., Arenas, A. & Rosvall, M. Identifying modular flows on multilayer networks reveals highly overlapping organization in interconnected systems. Phys. Rev. X 5, 011027 (2015).

**Have all data underlying the figures and results presented in the manuscript been provided?**

Reviewer #1: None

Reviewer #2: Yes

Reviewer #3: Yes

PLOS authors have the option to publish the peer review history of their article (what does this mean?). If published, this will include your full peer review and any attached files.

Reviewer #1: No

Reviewer #2: No

Reviewer #3: No
---

## [Decision Letter · Decision Letter 1]

25 May 2021

Dear Dr Baudot,

Thank you very much for submitting your manuscript "A Multi-Objective Genetic Algorithm to Find Active Modules in Multiplex Biological Networks" for consideration at PLOS Computational Biology. As with all papers reviewed by the journal, your manuscript was reviewed by members of the editorial board and by several independent reviewers. The reviewers appreciated the attention to an important topic. Based on the reviews, we are likely to accept this manuscript for publication, providing that you modify the manuscript according to the review recommendations.

Reviewer #3 still has concerns about parameter tuning and the dependence of the results on the parameters. I recommend that the authors expand the discussion section of the paper and convey these concerns to the readers. All algorithms --- and studies evaluating those algorithms --- have limitations and caveats, and the reviewer's comments should at least be presented as part of the study, even if they are not addressed by adding new results.

Feel free to contact me if I can be of assistance as you make the final changes to your manuscript.

Sincerely,

Paul Jensen

Guest Editor

PLOS Computational Biology

Mona Singh

Methods Editor

PLOS Computational Biology

[LINK]

Thank you for sharing your revised manuscript with us. Reviewer #3 still has concerns about parameter tuning and the dependence of the results on the parameters. I recommend that the authors expand the discussion section of the paper and convey these concerns to the readers. All algorithms --- and studies evaluating those algorithms --- have limitations and caveats, and the reviewer's comments should at least be presented as part of the study, even if they are not addressed by adding new results.

Please feel free to contact me if I can be of assistance as you make the final changes to your manuscript.

Reviewer's Responses to Questions

**Comments to the Authors:**

Reviewer #1: My major concerns with the work regarding comparison with other algorithms that have a similar goal and more robust testing of the proposed method, were addressed.

Reviewer #2: Minors:

Line 462: "the F_1" ==> "The F_1"

In the caption of Figure 4: "2014 cite34," Is it correct?

Check the last author's name in ref.[22], i.e. ",T..A.M.T. " ??

Reviewer #3: I appreciate the authors' effort in addressing my comments which have significantly improved the clarity of the manuscript. However, I feel a number of major issues have not been resolved.

Three major issues:

1. The evaluation of the proposed method lacks rigor and is not a fair comparison, as stated in the original comments. The proposed method, MOGAMUN, relies on a critical parameter, min_module_size. In fact, most modules identified by MOGAMUN has size very close to or slightly above min_module_size. For simulated data, this is a huge advantage for the proposed method as the authors can choose a min_module_size (15) close to the actual size (20). In contrast, the module size by the three competing methods (avg = 6, 640, and 6952, respectively) are VERY different from the actual size. I understand that the authors used default parameters from the competing methods; however, I believe the authors need to put some extra effort here to tune the parameters to make sure the module sizes are at least somewhat closer to the actual size for the simulated data. The size difference may be the most important reason that the competing methods had a much lower F-1 score and density (Fig 2, 3).

2. I am also concerned with the fact that most modules have size very similar to the predefined min_module_size, which means a relatively larger module will be reported as MANY smaller modules with significant overlaps. The reported modules in the supplementary file indeed show that. For example, in supplementary Fig 13, the 17 active modules seem to be subnetworks of two larger modules.

3. The authors claim that their method is the first to deal with multiplex networks. It may be true that no other method has utilized multiple networks the same way as the authors did, but the authors, in my opinion, failed to demonstrate the benefit of their approach of combining multiple networks. First, the simulated data has only a single network (which the authors has given a reason, so not the point to discuss here.). Then, on the real-world data, the authors compared two different ways of using multiple networks: (1) their original MAGAMUN, which is equivalent to precomputing a weighted merge of the networks and then trying to optimize subnetwork density on the weighted network, and (2) compute an UNWEIGTHED merge of the networks, which is then used by the competing algorithms and MAGAMUN to identify active modules. The results are presented as Sec 5 of the supplementary file and very briefly mentioned in manuscript. Similar to point 1 above, these results mostly shows that the module size from the other algorithms are either "too small" or "too large", which again, potentially could be addressed by the competing methods fairly easily. There is also not sufficient comparison of the two versions of MAGAMUN (with weighted or unweighted aggregated networks).

Two relatively minor issues:

4. Fig S16 does NOT show that the algorithm is insensitive to parameter tuning. A boxplot only shows that the distribution of node scores and densities are somewhat similar across different network parameters. It does not show that the actual network results are similar.

5. The authors have included a lot of suppl figs and tables, but many of them are not referred / discussed in the manuscript.

**Have the authors made all data and (if applicable) computational code underlying the findings in their manuscript fully available?**

Reviewer #1: None

Reviewer #2: Yes

Reviewer #3: None

PLOS authors have the option to publish the peer review history of their article (what does this mean?). If published, this will include your full peer review and any attached files.

Reviewer #1: No

Reviewer #2: No

Reviewer #3: No

Figure Files:

Data Requirements:

Reproducibility:

References:

---

## [Editor Report · Decision Letter 2]

9 Jul 2021

Dear Dr Baudot,

We are pleased to inform you that your manuscript 'A Multi-Objective Genetic Algorithm to Find Active Modules in Multiplex Biological Networks' has been provisionally accepted for publication in PLOS Computational Biology.

Best regards,

Paul Jensen

Guest Editor

PLOS Computational Biology

Mona Singh

Methods Editor

PLOS Computational Biology

---

## [Editor Report · Acceptance letter]

25 Aug 2021

PCOMPBIOL-D-20-01734R2 

A Multi-Objective Genetic Algorithm to Find Active Modules in Multiplex Biological Networks

Dear Dr Baudot,

I am pleased to inform you that your manuscript has been formally accepted for publication in PLOS Computational Biology. Your manuscript is now with our production department and you will be notified of the publication date in due course.

With kind regards,

Andrea Szabo
